# AutoCast++: Enhancing World Event Prediction with Zero-shot Ranking-based Context Retrieval

**Qi Yan**[1]* **Raihan Seraj**[2] **Jiawei He**[2] **Lili Meng**[3] **Tristan Sylvain**[2]
[1]University of British Columbia    [2]Borealis AI    [3]Independent Researcher
qi.yan@ece.ubc.ca   lilimeng1103@gmail.com
{raihan.seraj,jiawei.he,tristan.sylvain}@borealisai.com

## Abstract

Machine-based prediction of real-world events is garnering attention due to its potential for informed decision-making. Whereas traditional forecasting predominantly hinges on structured data like time-series, recent breakthroughs in language models enable predictions using unstructured text. In particular, (Zou et al., 2022) unveils AutoCast, a new benchmark that employs news articles for answering forecasting queries. Nevertheless, existing methods still trail behind human performance. The cornerstone of accurate forecasting, we argue, lies in identifying a concise, yet rich subset of news snippets from a vast corpus. With this motivation, we introduce AutoCast++, a zero-shot ranking-based context retrieval system, tailored to sift through expansive news document collections for event forecasting. Our approach first re-ranks articles based on zero-shot question-passage relevance, honing in on semantically pertinent news. Following this, the chosen articles are subjected to zero-shot summarization to attain succinct context. Leveraging a pre-trained language model, we conduct both the relevance evaluation and article summarization without needing domain-specific training. Notably, recent articles can sometimes be at odds with preceding ones due to new facts or unanticipated incidents, leading to fluctuating temporal dynamics. To tackle this, our re-ranking mechanism gives preference to more recent articles, and we further regularize the multi-passage representation learning to align with human forecaster responses made on different dates. Empirical results underscore marked improvements across multiple metrics, improving the performance for multiple-choice questions (MCQ) by 48% and true/false (TF) questions by up to 8%. Code is available at https://github.com/BorealisAI/Autocast-plus-plus.

## 1 Introduction

In the realm of machine learning research, event forecasting has emerged as an area of both theoretical intrigue and practical importance. The intersection of these two dimensions has placed event forecasting at the forefront of artificial intelligence inquiries, holding immense promise for real-world applications.

The initial focus of machine learning research in forecasting was predominantly on the prediction of *time-series* data (Shabani et al., 2023), a relatively straightforward task when compared to the complexity of real-world events. However, as the demand for more accurate forecasts in diverse domains has grown, the need to integrate data from beyond the structured time-series modality has become apparent. One such critical modality is the continuous stream of news articles, often presented in lengthy textual formats. In the pursuit of predicting future events, the analysis and interpretation of news articles have become central to the endeavor.

Recent advancements in this field, exemplified by initiatives like the Autocast dataset, have demonstrated the potential of utilizing news articles to provide probabilistic estimates of real-world events. Nevertheless, it is evident that the field of event forecasting through machine learning is still in its

---

*Work performed while interning at Borealis AI.

early stages. Despite promising results, these methods have yet to reach the level of proficiency exhibited by human forecasters. A considerable gap exists between the theoretical potential and the practical feasibility of machine learning-based event forecasting.

In this research paper, we posit that progress in event forecasting methods relies on addressing three fundamental questions concerning the analysis of news articles. These questions form the core of our investigation:

- How can the selection of the most relevant news articles for event forecasting be improved?
- How can the processing of selected news articles be optimized to extract information effectively?
- How can the learning dynamics be refined to enable efficient model training and leverage the enriched information from relevant news articles?

This paper introduces a series of novel components designed to provide answers to these critical questions, ultimately enhancing the entire event forecasting process. Our contributions encompass the following key aspects:

- **Task-Aligned Retrieval Module:** We present a task-aligned retrieval module that aligns the information retrieval process with the central task of answering critical questions, thereby increasing the relevance of selected news articles.
- **Enhanced Neural Article Reader:** We propose an improved neural article reader, enhanced through unsupervised distillation techniques, including summarization. This augmentation enables the model to delve deeper into news articles, efficiently extracting pertinent insights.
- **Human-Aligned Loss Function:** We introduce a novel human-aligned loss function that considers human preferences and judgments, bridging the gap between machine learning models and human intuition.

Through thorough testing and comprehensive evaluation, our study demonstrates that our proposed components play a vital role, as supported by ablation studies, and also contribute to substantial performance improvements. Our findings indicate that the proposed technique significantly enhances baseline models across various metrics, resulting in a noteworthy 48% boost in accuracy for multiple-choice questions (MCQ), an appreciable 8% improvement for true/false (TF) questions, and a substantial 19% enhancement for numerical predictions. These results emphasize the progress we have made in advancing event forecasting through machine learning. Our research represents a significant step toward realizing the full potential of machine learning in event forecasting, aligning theoretical exploration with practical applications in the domain of AI-driven predictions.

## 2 RELATED WORK

**Event Forecasting.** Using large language models (LLMs) to make predictions is an increasingly active area of research (Lopez-Lira & Tang, 2023). One of the pioneering datasets in event forecasting is ForecastQA (Jin et al., 2021). However, as highlighted by (Zou et al., 2022), this dataset has shortcomings, such as the presence of non-sensical questions or potential data leakage in retrospective human forecasting labels. To address this, (Zou et al., 2022) introduced a large-scale, balanced dataset tailored for event forecasting, which serves as the primary evaluation setting in our study.

**Information Retrieval.** Information retrieval (Guo et al., 2016; Mitra & Craswell, 2017; Zhao et al., 2022) aims to extract pertinent information in response to specific queries from a vast textual repository. Earlier methods such as BM-25r (Robertson et al., 1995) relied on a bag-of-words retrieval function. More recently, there has been a growing interest in leveraging information retrieval to enhance the question-answering capabilities of LLMs (Lewis et al., 2020; Shuster et al., 2021). In a parallel vein, LLMs can also be directly employed for the purpose of information retrieval (Zhu et al., 2023).

**LLMs for Text Summarization.** Recent studies (Liu et al., 2023; Zhang et al., 2023b) have highlighted the capabilities of large language models (LLMs) in text summarization. Notably, the GPT series (Brown et al., 2020; OpenAI, 2023) and the T5 model (Raffel et al., 2020) have emerged as

leaders in this domain, showcasing the advancements and versatility of LLMs in condensing textual content. In addition to these generalist architectures, some approaches such as (Liu et al., 2022; Liu & Liu, 2021) are dedicated to summarization.

**Open-domain Question Answering.** Open domain question answering (OpenQA) focuses on generating factual answers from expansive document collections without explicit evidence pointers (Zhang et al., 2023a). Unlike many QA tasks, OpenQA relies solely on the posed question, devoid of any associated documents that might house the answer (Sachan et al., 2021). Typical OpenQA methodologies incorporate both retriever and reader mechanisms. The retriever's role is to pinpoint a concise set of documents to assist the reader (Chen et al., 2017), and subsequently, the reader is calibrated to produce a succinct answer to the query (Izacard & Grave, 2021). The Unsupervised Passage Re-ranker (UPR) stands as an exemplar in this field, emphasizing unsupervised passage re-ranking for open-domain retrieval (Sachan et al., 2022).

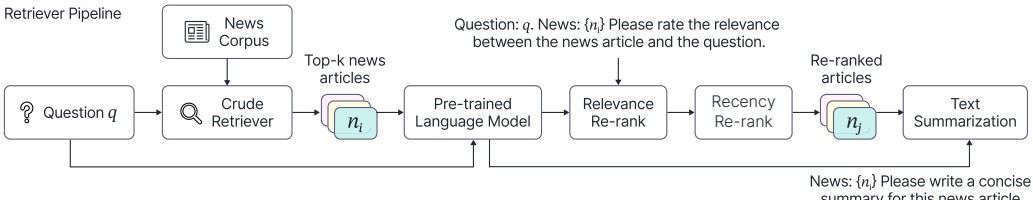

(a) Retriever model with zero-shot re-ranking and text summarization.

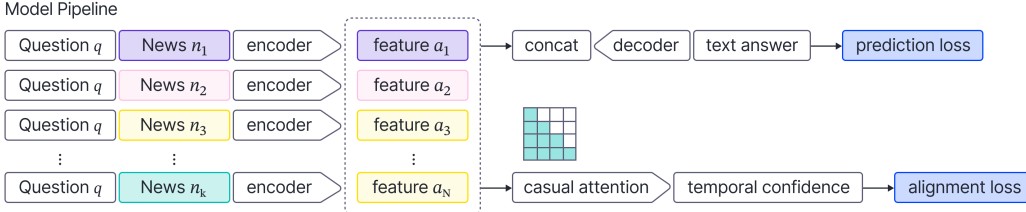

(b) Reader model based on Fusion-in-Decoder (FiD) (Izacard & Grave, 2021).

Figure 1: Illustration of the Autocast++ components. **Top:** For each question, the retriever employs zero-shot relevance re-ranking and recency re-ranking to pinpoint relevant news articles from a large corpus, subsequently using unsupervised text summarization to establish a concise context. **Bottom:** Our FiD-based reader utilizes the generative decoder for predicting event outcomes. We also introduce an auxiliary alignment loss to synchronize with the responses of human forecasters.

## 3 METHOD

Event forecasting through news articles demands three specific capabilities: accurately pinpointing relevant articles, adept processing of these articles, and ensuring the learning dynamics optimally harness the acquired information. To enhance these capacities, we present a zero-shot ranking-based retriever-reader model that emphasizes effective context retrieval and alignment with human forecasters. A comprehensive depiction of our architecture is illustrated in Figure 1. Each of the following subsections provides a key improvement that we introduce to address limitations of past approaches in the three areas listed above.

### 3.1 PRELIMINARY

Our model is grounded in the retriever-reader framework. For a given query $q$ from the question set $\mathcal{Q}$, the retriever module strives to select a subset of pertinent articles, $\mathcal{N}_q$, from the news passages database represented by $\mathcal{D} = \{n_{i_1}, n_{i_2}, \cdots, n_{i_M}\}$[1]. The articles retrieved in relation to question $q$ are represented as $\mathcal{N}_q = \{n_1, n_2, \cdots, n_K\}$, where $\mathcal{N}_q \in \mathcal{D}$ is the retrieved subset of news

---

[1]We represent the canonical order of news articles in the dataset $\mathcal{D}$ using the notation $\{i_1, i_2, \cdots, i_M\}$. This aims to differentiate from the article index in the retrieval set $\mathcal{N}_q$.

articles. Each query $q$ includes the question text, potential answer choices, start and end dates of the question, and the type of question. Each news article $n_i$ within the retrieval set $\mathcal{N}_q$ comprises a published date, title, text, and a relevance score relative to the query $q$. This paired data $(q, \mathcal{N}_q)$ is then input into the reader network to forecast an event outcome $o$. The event outcome $o$ is defined as a discrete variable corresponding to one of the allowable answers. For example, in the case of True/False questions, the outcome is represented as $o \in \{\texttt{True}, \texttt{False}\}$. We discretize responses for continuous-valued numerical questions to make this condition hold, which will be further detailed later on. Our reader employs an encoder-decoder transformer, designed to produce answers for the forecasting question using the generative model $p(o|q, \mathcal{N}_q; \Theta)$, where $\Theta$ means the reader parameters. The reader objective is to maximize the likelihood of generating the actual outcome $o_{\text{gt}}$: $\arg\max_\Theta p(o = o_{\text{gt}}|q, \mathcal{N}_q; \Theta)$. Specifically, our reader model adopts the Fusion-in-Decoder (FiD) (Izacard & Grave, 2021) on top of T5 (Raffel et al., 2020).

Significantly, the Autocast dataset provides intermediate forecasting results from human experts for every query. These forecasts are documented at various timestamps spanning the start and end of the question period. The notation $p_h(o|q, t)$ represents the probabilistic human prediction[2] made at the timestamp $t$ for the specific question $q$. Specifically, $p_h(o = o_{\text{gt}}|q, t)$ denotes the accuracy of the human forecasters. We employ such human feedback to steer both our retriever and reader models towards effective forecasting, which will be detailed in the following sections.

## 3.2 TASK-ALIGNED RETRIEVAL MODULE

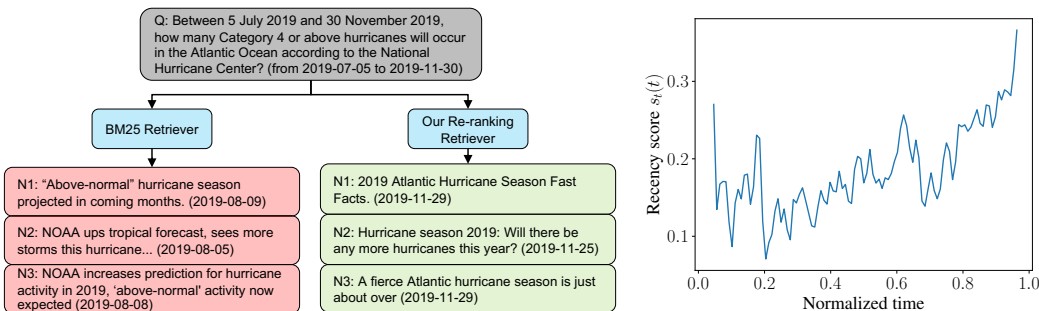

Figure 2: **Left**: retrieval results of BM25 and our re-ranking retriever. Given a query seeking information about a hurricane until its expiration date, the BM25 identifies articles based on lexical similarity. However, these articles, closely aligned with the query start date, lack depth for a retrospective answer. In contrast, our re-ranking retriever focuses on more recent, relevant passages to effectively address the query. **Right**: Visualization of news recency score $s_t(t)$. As the query expiry date nears, human forecaster accuracy typically increases more rapidly. This score is a statistical measure capturing general patterns across the whole dataset.

We posit the existence of a preliminary retriever (specifically BM25 as adopted in previous research), which consumes the question $q$ and the document set $\mathcal{D}$ to approximate the $K$ most relevant articles $\mathcal{N}_q$. While many forecasting studies (Zou et al., 2022; Jin et al., 2021) depend on BM25, it often falls short in differentiating between articles that directly address the question and those that merely focus on a related theme. As a result, we aim to improve the retrieval module.

**Zero-shot News Relevance Assessment.** From the top-$K$ news articles $\mathcal{N}_q$ identified by the initial retriever, our goal is to derive robust quantitative measures on the relevance between questions and news to enhance article retrieval. We subsequently re-rank these articles based on the relevance score $s_r(n_i, q), \forall n_i \in \mathcal{N}_q$, obtained in a zero-shot manner. This means we forgo the need for task-specific training data, leveraging a pre-trained LLM for ease of implementation.

Estimating the inherent value of $s_r(n_i, q)$ is admittedly intricate (Sachan et al., 2021; 2022), given the absence of definitive ground-truth. To bring clarity to relevance-based re-ranking, we design

---

[2]Adhering to the conventions established in the Autocast dataset as detailed in (Zou et al., 2022), we treat the aggregated human crowd judgment as a probabilistic measure.

binning-based relevance score estimation for $s_r(\boldsymbol{n}_i, \boldsymbol{q})$. Instead of estimating the continuous-valued score directly, we first assess a discrete relevance label $g$ that corresponds to up to $G$ bins, evenly distributed in the range of 0 to 1. The LLM assessment output g represents the relevance, which is quantified on a scale from 1 to $G$:

$$s_r(\boldsymbol{n}_i, \boldsymbol{q}) \approx \frac{\mathbb{E}[\text{g}]}{G-1}, \text{g} \sim p(\text{g}|\boldsymbol{n}_i, \boldsymbol{q}; \Psi). \tag{1}$$

Here, $\Psi$ denotes the parameters of the pretrained LLM, and $\text{g} \in \{0, 1, \cdots, G-1\}$ represents the discrete relevance metric. We utilize the LLM in a zero-shot manner, refraining from fine-tuning on the task-specific dataset. To the question and article tokens, we append a straightforward language prompt such as: "*Please rate the relevance between the news article and the question on a scale of 0 to 4, with 4 being the most relevant.*", where we set $G = 5$. We run LLM inference multiple times (denoted by $l$) to estimate $\mathbb{E}[\text{g}] = 1/l \sum_{i=1}^{l} g_i$ and evaluate $s_r(\boldsymbol{n}_i, \boldsymbol{q})$.

**Accounting for News Recency based on Human Feedback.** The timeliness of context passages is pivotal in determining their usefulness. For instance, when addressing forecasting queries about natural disasters, news reports closer to the question ending date often hold more value compared to early-stage reports. The ever-changing dynamics of the real world profoundly impact the favored responses to forecasting queries. Thus, solely considering the content-based news-question relevance $s_r(\boldsymbol{n}_i, \boldsymbol{q})$ is inadequate for effective retrieval. To address this, we leverage human-feedback statistics to gauge temporal truthfulness and prioritize more recent news.

We arrange the articles in $\mathcal{N}_{\boldsymbol{q}}$ chronologically as $\boldsymbol{n}_{\tau_1}, \boldsymbol{n}_{\tau_2}, \cdots, \boldsymbol{n}_{\tau_K}$, where the publication date for news $\boldsymbol{n}_{\tau_i}$ is $t_i$ and $t_1 \leq t_2 \cdots \leq t_K$, with $t_K$ being the most recent date. We postulate that the recency score $s_t(\boldsymbol{n}_{\tau_K}, \boldsymbol{q})$ for news dated $t_K$ correlates with the temporal performance enhancement observed in human forecasters:

$$s_t(\boldsymbol{n}_{\tau_K}, \boldsymbol{q}) \approx \frac{p_h(\boldsymbol{o} = \boldsymbol{o}_{\text{gt}}|\boldsymbol{q}, t_K) - p_h(\boldsymbol{o} = \boldsymbol{o}_{\text{gt}}|\boldsymbol{q}, t_{K-1})}{t_K - t_{K-1}}. \tag{2}$$

Human forecasters provide responses at time $t_K$ using information available up to that point, encompassing the articles $\boldsymbol{n}_{\tau \leq K}$. We assess $s_t(\boldsymbol{n}_{\tau_K}, \boldsymbol{q})$ by examining the variation in human forecaster accuracy, averaged over the time gaps between two successive articles. To derive temporal dynamics that are agnostic to specific query-news pairings, we calculate the expectation across the empirical question distribution $\boldsymbol{q} \in \mathcal{Q}$ and its top-$K$ news articles $\mathcal{N}_{\boldsymbol{q}}$ distribution:

$$s_t(t) \approx \mathbb{E}_{\boldsymbol{q} \in \mathcal{Q}} \mathbb{E}_{\boldsymbol{n}_t \in \mathcal{N}_{\boldsymbol{q}}} [s_t(\boldsymbol{n}_t, \boldsymbol{q})]. \tag{3}$$

To cope with queries spanning multiple years, we use normalized time $t$ relative to the question start and expiry dates instead of absolute time. Putting things together, we combine the scores defined in Eq. (1) and Eq. (3) to obtain the retrieval score:

$$s(\boldsymbol{n}_i, \boldsymbol{q}) = \underbrace{s_r(\boldsymbol{n}_i, \boldsymbol{q})}_{\text{content relevance}} \cdot \underbrace{s_t(t_{\boldsymbol{n}_i})}_{\text{recency}}, \tag{4}$$

where $t_{\boldsymbol{n}_i}$ denotes the normalized time of news $\boldsymbol{n}_i$. After computing the retrieval score $s(\boldsymbol{n}_i, \boldsymbol{q})$ for each $\boldsymbol{q} \in \mathcal{Q}$ and $\boldsymbol{n}_i \in \mathcal{N}_{\boldsymbol{q}}$, we reorder the articles in $\mathcal{N}_{\boldsymbol{q}}$ and select the top-$N$ out of the $K$ passages. Fig. 2 shows the effectiveness of our retriever against the vanilla BM25 and the recency score $s_t(t)$.

### 3.3 Neural Reader with Unsupervised Distillation

Following the retrieval, our reader processes the question $\boldsymbol{q}$ along with its associated articles $\mathcal{N}_{\boldsymbol{q}}$ to formulate answers to the forecasting inquiry. To refine context clarity, we advocate for an unsupervised distillation on the retrieved news articles, achieved through abstractive summarization. These distilled articles are then fed to the FiD reader for acquiring textual representations and for learning to generate forecasting-related answers.

**Zero-shot News Summarization.** News articles often encompass lengthy segments, including interviews and analyses, which might not directly provide factual insights. Extracting significant information from these potentially relevant sections poses a challenge. As a solution, rather than processing the raw articles $\mathcal{N}_{\boldsymbol{q}}$ in the reader network, we turn to an abstractive summarization tool to produce succinct and pertinent summaries. Our method leverages a pre-trained LLM, prompting it with "*Please write a concise summary for this news article.*" to carry out text summarization without the necessity of supervised signals.

**Fusion-in-Decoder Reader Architecture.** In the reader model, we combine the tokens of a question $\boldsymbol{q}$ with the tokens from a single news article sourced from its top-$N$ retrieval results, $\mathcal{N}_{\boldsymbol{q}}$. Each question-news pair undergoes the prepending process independently, denoted by $\boldsymbol{x}_i = \texttt{prepend}(\boldsymbol{q}, \boldsymbol{n}_i)$. We consider temporal data, including the start and end dates of queries and the publication dates of news in the prepending step. Please see the appendix for details.

Subsequently, the T5 encoder $f_e$ processes the concatenated tokens $\{\boldsymbol{x}_i\}_{i=1}^N$, resulting in a textual representation for the sequence:

$$\forall i \in [N], \boldsymbol{z}_i = f_e(\boldsymbol{x}_i; \Theta); \boldsymbol{X}_e = \texttt{concat}(\boldsymbol{z}_1, \boldsymbol{z}_2, \cdots, \boldsymbol{z}_N). \tag{5}$$

Thereafter, the T5 decoder $f_d$ derives the answer for the forecasting question, employing both cross-attention and causal self-attention mechanisms. These mechanisms account for the tokens in $\boldsymbol{X}_e$ and the tokens generated previously, respectively. The underlying principle of concatenating representations from diverse documents is to provide the decoder with a holistic view of the information. The answer generation process is modeled by the autoregressive decoder $p(\boldsymbol{o}|\boldsymbol{q}, \mathcal{N}_{\boldsymbol{q}}; \Theta)$.

### 3.4 Training Objectives with Human Alignment Loss

Unlike prior research on generative question answering, our focus lies on real-world event prediction tasks which necessitate concise and definitive predictions rather than elaborate textual responses. Notably, our model is designed to predict numerical values, an area where traditional language models often falter (Lightman et al., 2023). Moreover, our model acknowledges the temporal dynamics intrinsic to real-world events, as evidenced by the sentiment shifts observed in news articles over time. To accommodate these unique challenges, we introduce multiple modifications to the conventional autoregressive generative decoder loss, aiming to enhance performance.

**Binning Numerical Questions.** The generative decoder $p(\boldsymbol{o}|\boldsymbol{q}, \mathcal{N}_{\boldsymbol{q}}; \Theta)$ inherently struggles with direct numerical value prediction. The earlier baseline (Zou et al., 2022) attempted to address this by integrating a linear layer based on the attention layer features for regression. We propose an alternative approach that retains the objective in the textual token space and avoids introducing such regression layers. Let $\boldsymbol{o}_{\text{num}} \in \mathbb{R}$ be the continuous-valued nuemrical outcome, which is categorized into $R$ groups: $\boldsymbol{o}'_{\text{num}} \leftarrow \lceil \frac{\boldsymbol{o}_{\text{num}}}{(\boldsymbol{o}_{\text{num}}^{\max} - \boldsymbol{o}_{\text{num}}^{\min})/R} \rceil \in \{1, \cdots, R\}$. During training, we use the discrete $\boldsymbol{o}'_{\text{num}}$ to serve as a proxy training target. During inference, to revert to the numerical value space, we employ the median value of each category: $\boldsymbol{o}_{\text{num}} \leftarrow \boldsymbol{o}'_{\text{num}} \frac{(\boldsymbol{o}_{\text{num}}^{\max} - \boldsymbol{o}_{\text{num}}^{\min})}{2R}$. In the Autocast dataset, with values given as $\boldsymbol{o}_{\text{num}}^{\max} = 1$ and $\boldsymbol{o}_{\text{num}}^{\min} = 0$, this discretization process is substantially simplified.

**Human Annotation Alignment Loss.** The Autocast dataset offers intermediate probabilistic responses from human forecasters, denoted by $p_h(\boldsymbol{o}|\boldsymbol{q}, t)$, gathered over various dates. Utilizing these labels, we harmonize the T5 encoder text representations $\{\boldsymbol{z}_i\}_i$ with the beliefs held by human forecasters. Consider the merged question-news token sequences $\{\boldsymbol{x}_i\}_i$ to be arranged chronologically, following $t_{\boldsymbol{x}_i} \le t_{\boldsymbol{x}_{i+1}} \cdots$. We employ a self-attention mechanism integrated with a causal mask, founded upon the text features $\{\boldsymbol{z}_i\}_i$. This layer aims to infer the contextual confidence up to time instant $t$: $p(u_t|\boldsymbol{z}_{\le t}; \Phi)$. Here, $u_t \in [0, 1]$ symbolizes the confidence, while $\Phi$ denotes the self-attention layer parameters. Our objective is to synchronize the inferred confidence with the accuracy of the human forecaster, as represented by $p_h(\boldsymbol{o}_{\text{gt}}|\boldsymbol{q}, t)$, thereby regularizing the learning of the text representation. Consequently, the overall training goal is to minimize the following loss:

$$L = -\underbrace{\log p(\boldsymbol{o}|\boldsymbol{q}, \mathcal{N}_{\boldsymbol{q}}; \Theta)}_{\text{decoder loss}} - \lambda \frac{1}{N} \sum_{t=1}^N \underbrace{D_{KL}\left(p_h(\boldsymbol{o}_{\text{gt}}|\boldsymbol{q}, t) \| p(u_t|\boldsymbol{z}_{\le t}; \Phi)\right)}_{\text{alignment loss}}, \tag{6}$$

where $\lambda$ is a weighting coefficient. We use cross-entropy loss for both terms in implementation.

## 4 Experiments

### 4.1 Dataset and Metrics

We assess our model on the **Autocast** dataset (Zou et al., 2022), a benchmark comprising a diverse set of annotated questions, including True/False (TF), Multiple-choice Questions (MCQ), and numerical predictions. These span various domains, such as economics, politics, and technology. We

Table 1: The number of questions in Autocast dataset, grouped by question type.

| Question Type | Train | Test | Total |
|---|---|---|---|
| T/F | 3187 | 775 | 3962 |
| MCQ | 753 | 176 | 929 |
| Numerical | 471 | 341 | 812 |

incorporate news articles from the Common Crawl corpus[3] spanning 2016 to 2022 for retrieval purpose. For performance evaluation, we use accuracy metrics for T/F and MCQ questions and absolute error for numerical prediction questions. The dataset is partitioned with a cut-off in mid-2021 and questions in the test set span from mid-2021 to mid-2022.

## 4.2 Experimental setup

**Implementation Details.** For relevance evaluation and text summarization, we utilize the pre-trained GPT-3 model (Brown et al., 2020). We initially retrieve $K = 50$ news articles using BM25 and proceed with our re-ranking process to select $N = 10$ unless otherwise specified. The reweighting coefficient $\lambda$ in Eq. (6) is fixed at 0.1. For most of our experiments, especially hyper-parameter and architecture optimization, we employ LoRA (Hu et al., 2022), fine-tuning it over the pre-trained T5 backbone. More results can be found in the appendix.

**Baselines.** Following the Autocast (Zou et al., 2022) benchmark, we incorporate these baselines: 1) UnifiedQA-v2 (Khashabi et al., 2022), 2) T5 (Raffel et al., 2020), 3) FiD Static (Izacard & Grave, 2021), and 4) FiD Temporal built upon GPT-2 (Radford et al., 2019). UnifiedQA-v2 and T5 are no-retrieval models, meaning they do not extract information from news articles. Specifically, UnifiedQA-v2 uses zero-shot prompting for answers, while T5 undergoes fine-tuning on the training data. In contrast, the FiD Static (T5-based) retrieves the top 10 news articles throughout the period, whereas the FiD Temporal (GPT-2 with T5 encoder) sources the top article daily. Although newer LLMs might demonstrate enhanced capabilities and potentially outperform in forecasting question answering, we avoid using or comparing them to avoid potential contamination from their more recent training data. It is important to highlight that a model pre-trained on data post-mid-2021 (training/test cut-off time) will not genuinely replicate the real-world forecasting condition.

## 4.3 Main Results

Table 2 showcases the primary results of our approach alongside the baselines highlighted in (Zou et al., 2022). It's pivotal to note that our reader's architectural framework aligns with FiD Static, with both leveraging pre-trained T5 weights. Notably, our proposed model eclipses the FiD Static baseline in multiple-choice questions (MCQ) by a margin of 48% (43.8 versus 29.6), true/false (TF) questions by 8% (66.7 versus 62.0) and numeircal predicion by 19% (19.8 vs 24.5) within the 0.2B model size category. On the other hand, numerical predictions remain a challenging task. For the 0.2B and 0.8B model sizes, our models achieve the best results in the class, leading by a margin of approximately 1%.

Furthermore, our more compact models, such as the 0.2B and 0.8B variants, already surpass larger baselines like the FiD Static with 2.8B parameters. This suggests that model size may not be the primary determinant for forecasting question answering. In a similar vein, we observe that the performance boost from utilizing more extensive models in our scenario isn't as remarkable. For instance, our 2.8B parameter model marginally outperforms the 0.2B model for T/F and MCQ, yet delivers identical outcomes for numerical predictions. While the 4.3B parameter FiD Temporal baseline achieves the best performance in numerical prediction, it is significantly larger and more computationally intensive to train. This is because it processes news articles for each day within the active period of a question, which averages 130 days in the Autocast dataset. Consequently, its retriever deals with a considerably larger volume of news articles. However, FiD Temporal does not perform satisfactorily on T/F and MCQ questions in contrast to FiD Static or our method. This

---

[3]Common Crawl - Open Repository of Web Crawl Data, `https://commoncrawl.org/`

Table 2: Performance of our approach using three different model sizes. We take the baseline results from (Zou et al., 2022). Across various model sizes, our method demonstrates a substantial performance boost, notably for smaller models. It significantly surpasses the retrieval baselines of FiD Static and FiD Temporal, underscoring the efficacy of our proposed retriever-reader model.

| Model | Full Context Length | Model Size | T/F↑ | MCQ↑ | Num.↓ |
|---|---|---|---|---|---|
| **Small-size Models** | | | | | |
| UnifiedQA (Khashabi et al., 2022) | n/a | 0.2B | 45.4 | 23.5 | 34.5 |
| T5 (Raffel et al., 2020) | n/a | 0.2B | 61.3 | 24.0 | 20.5 |
| FiD Static (Zou et al., 2022) | 10 | 0.2B | 62.0 | 29.6 | 24.5 |
| FiD Temporal (Zou et al., 2022) | query-specific | 0.6B | 62.0 | 33.5 | 23.9 |
| **Autocast++ (ours)** | 10 | 0.2B | 66.7 | 43.8 | 19.8 |
| **Middle-size Models** | | | | | |
| UnifiedQA (Khashabi et al., 2022) | n/a | 0.8B | 48.2 | 23.5 | 34.5 |
| T5 (Raffel et al., 2020) | n/a | 0.8B | 60.0 | 29.1 | 21.7 |
| FiD Static (Zou et al., 2022) | 10 | 0.8B | 64.1 | 32.4 | 21.8 |
| FiD Temporal (Zou et al., 2022) | query-specific | 1.5B | 63.8 | 32.4 | 21.0 |
| **Autocast++ (ours)** | 10 | 0.8B | 67.3 | 44.0 | 19.9 |
| **Large-size Models** | | | | | |
| UnifiedQA (Khashabi et al., 2022) | n/a | 2.8B | 54.9 | 25.1 | 34.5 |
| T5 (Raffel et al., 2020) | n/a | 2.8B | 60.0 | 26.8 | 21.9 |
| FiD Static (Zou et al., 2022) | 10 | 2.8B | 65.4 | 35.8 | 19.9 |
| FiD Temporal (Zou et al., 2022) | query-specific | 4.3B | 62.9 | 36.9 | **19.5** |
| **Autocast++ (ours)** | 10 | 2.8B | **67.9** | **44.1** | 19.8 |

highlights the critical importance of effectively extracting relevant information from news articles, rather than indiscriminately processing all available data.

## 4.4 ABLATION STUDIES

To substantiate the effectiveness of our introduced components, we perform ablation studies particularly in comparison to the FiD Static model. In the context of retrieval, the FiD Static integrates the BM25 retriever enhanced with cross-encoder reranking. The architecture of both the FiD Static and our model's reader networks align until the self-attention module tailored for the human alignment loss and the training loss for binning numerical question. By implementing modifications progressively, we can evolve from the FiD Static to our design, facilitating an evaluation of the incremental performance benefits of each component. In Table 3, we detail the ablation experiments utilizing the 0.2B model size, with the baseline model being the vanilla FiD Static. To ease exposition, we name FiD Static with numerical question binning FiD Static*. We employ distinct markers to clearly differentiate between various ablations: ① represents experiments on LLM-enhanced components, ② denotes experiments on numerical question binning, and ③ indicates experiments on alignment loss.

Table 3: Effects of different retriever-reader components on performance.

| Model | Retriever Component | | Reader Component | | | Metrics | | |
|---|---|---|---|---|---|---|---|---|
| | Zero-shot Relevance Re-rank | Recency Re-rank | Zero-shot Sum. | Binning Num. Questions | Alignment Loss | T/F↑ | MCQ↑ | Num↓ |
| FiD Static | ✗ | ✗ | ✗ | ✗ | ✗ | 62.0 | 29.6 | 24.5 |
| ① FiD Static + rel. re-rank & sum. | ✓ | ✗ | ✓ | ✗ | ✗ | 65.5 | 42.1 | 21.3 |
| ③ FiD Static + alignment | ✗ | ✗ | ✗ | ✗ | ✓ | 62.5 | 31.2 | 23.5 |
| ② FiD Static* | ✗ | ✗ | ✗ | ✓ | ✗ | 62.1 | 29.6 | 23.2 |
| ① FiD Static* + sum. | ✗ | ✗ | ✓ | ✓ | ✗ | 65.7 | 42.3 | 19.8 |
| ① FiD Static* + rel. re-rank | ✓ | ✗ | ✗ | ✓ | ✗ | 64.7 | 39.8 | 21.2 |
| ① FiD Static* + full retriever | ✓ | ✓ | ✗ | ✓ | ✗ | 65.8 | 42.4 | 20.2 |
| ② Autocast++ w/o binning | ✓ | ✓ | ✓ | ✗ | ✓ | 66.7 | 43.6 | 21.0 |
| ③ Autocast++ w/o alignment | ✓ | ✓ | ✓ | ✓ | ✗ | 66.7 | 43.5 | 19.9 |
| Autocast++ full model | ✓ | ✓ | ✓ | ✓ | ✓ | **66.7** | **43.8** | **19.8** |

①  **Effects of LLM Enhancement.**  The zero-shot relevance re-ranking and text summarization techniques, facilitated by the pre-trained LLM, stand out as the most effective among our proposed methods.  Implementing these techniques in FiD Static or FiD Static* baselines significantly improves performance across various question types, with a marked enhancement in MCQ accuracy, elevating their performance to nearly match that of the full Autocast++ model.

②  **Effects of Binnning Numerical Question.**  We propose binning numerical questions to transform the regression task in a continuous space into a classification problem.  This approach allows us to apply the same cross-entropy loss used for T/F and MCQ. Empirically, we observe that this method generally enhances the performance of numerical questions (*e.g.*, in FiD Static or the full Autocast++ model) without adversely affecting other question types.  Although its impact on overall metrics is less significant, it complements the LLM enhancement components well.

③  **Effects of Alignment Loss.**  The alignment loss leverages human crowd forecasting results to regulate the text encoder representations, thereby simulating the progression of information and knowledge over time.  This alignment loss is particularly beneficial for the basic FiD Static baseline, which lacks any LLM-enhanced components.  However, in the full Autocast++ model, where relevance re-ranking and text summarization are employed, the alignment loss appears to have a diminished role in enhancing performance.

### 4.5  QUALITATIVE EXAMPLES

---

**Question:** Between 5 July 2019 and 30 November 2019, how many Category 4 or above hurricanes will occur in the Atlantic Ocean according to the National Hurricane Center? (from 2019-07-05 to 2019-11-30) **Choices**: A: 0, B: 1, C: 2, D: 3, E: 4 or more. **Answer**: C.
**Our retriever:**
N1: 2019 Atlantic Hurricane Season Fast Facts. (2019-11-29) - Text omitted
N2: Hurricane season 2019: Will there be any more hurricanes this year? (2019-11-25) - Text omitted
N3: A fierce Atlantic hurricane season is just about over (2019-11-29) Text: The 2019 hurricane season, which ended recently, was marked by significant and historic destruction in various regions. It produced a total of six hurricanes, with three reaching major hurricane status (winds of at least 111 mph). Additionally, there were 12 tropical or subtropical storms and two tropical depressions. Notably, this season included two Category 5 hurricanes: Dorian and Lorenzo. This marks the fourth consecutive above-normal Atlantic hurricane season, a pattern last observed from 1998 to 2001. In the Gulf of Mexico, five tropical cyclones formed, tying a record set in 2003 and 1957. Three of these storms—Barry, Imelda, and Nestor—made landfall in the United States. Several factors contributed to the favorable conditions for hurricanes during this season. These factors included a multi-decade cycle of high hurricane activity since 1995, a stronger West African monsoon, warmer Atlantic waters, and weak wind shear, which is crosswinds that can hinder hurricane formation. Hurricane Dorian, one of the most significant storms of the season, brought devastation to the Bahamas with near-record strength, reaching winds of 185 mph. It continued to impact the east coast and Nova Scotia. Hurricane Dorian's strength ties it with three other historic hurricanes as the second strongest hurricane on record in the Atlantic basin in terms of wind speed. Another Category 5 hurricane, Lorenzo, caused less destruction than Dorian as it quickly veered north. However, it did sink a French tugboat and brought strong winds to Ireland, resulting in flooding and power outages. The 2019 hurricane season officially runs from June 1 to November 30, though storms can occur outside these dates. As the season has come to an end, the Atlantic Ocean is currently free of tropical storm activity, and the National Hurricane Center does not anticipate any storms in the Atlantic, Gulf of Mexico, or Caribbean Sea in the remaining days of the season.

---

Figure 3: An example of our proposed re-ranking retrieval with text summarization.

In Fig. 3, we present an expanded view of qualitative results corresponding to the retriever comparison depicted in Fig. 2. Specifically, we focus on the news context after summarization for the most informative news article, which we identify as N3 (news with third highest relevance score). With the key information highlighted in orange, the model can readily make an informed inference about the answer, which is C (2 Category 4 or above hurricanes during this period). The integration of pretrained LLM into the retrieval and summarization processes significantly improves the extraction of pertinent textual information while filtering out less relevant content. This enhancement proves particularly beneficial, as demonstrated in our ablation experiment labeled as ①.

## 5  CONCLUSION

In this research paper, we introduced three key components that significantly enhance the event forecasting process: 1) task-aligned retrieval module, 2) enhanced neural article reader, and 3) human-aligned loss function. Our work has significantly boosted accuracy in various question types, such as a 48% improvement in multiple-choice questions, an 8% enhancement in true/false questions, and a substantial 19% increase in numerical predictions. These results highlight our progress in advancing event forecasting through machine learning, bridging the gap between theory and practical AI-driven applications. In conclusion, our research represents a significant step forward in the field of event forecasting through machine learning. As the field continues to evolve, we anticipate further breakthroughs and innovations that will shape the future of event forecasting and its practical applications.

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

# A  IMPLEMENTATION DETAILS

**Question Appending.**   In FiD, we concatenate the question $q$ and its top-$N$ retrieved articles $\mathcal{N}_q$ to encode the input as a whole:

$$q' = q \, [\texttt{SEP}] \, \texttt{date}(q) \, [\texttt{SEP}] \, \texttt{choices}(q),$$
$$n'_i = \texttt{title}(n_i) \, [\texttt{SEP}] \, \texttt{date}(n_i) \, [\texttt{SEP}] \, n_i,$$
$$x_i = [\texttt{CLS}] \, q' \, [\texttt{SEP}] \, n'_i \, [\texttt{SEP}],$$

where $[\texttt{CLS}]$ is the beginning of the sequence, and $[\texttt{SEP}]$ is the separator token.

To provide a comprehensive description of both the query and the context, the question is augmented with its starting and ending dates, along with its allowable choices. Also, the beginning of news article is prefixed with its title and date. Notably, the notation $n_i$ is repurposed to denote the summarized article.

# B  ADDITIONAL EXPERIMENTAL RESULTS

## B.1  REPRODUCING RESULTS WITH LoRA

In this subsection, we demonstrate that LoRA (Hu et al., 2022) enables us to reproduce the outcomes presented in the original paper (Zou et al., 2022) while maintaining performance at a virtually indistinguishable level. We use LoRA to explore different model design choices and carry out hyperparameter search to save time. We carry out fine-tuning without LoRA for final experiments.

Table 4: Effects of LoRA on FiD Static model performance.

| Model | Model Size | T/F↑ | MCQ↑ | Num↓ |
|---|---|---|---|---|
| FiD Static (Zou et al., 2022) | 0.2B | 62.0 | 29.6 | 24.5 |
| FiD Static (Zou et al., 2022) | 0.8B | 64.1 | 32.4 | 21.8 |
| FiD Static (Zou et al., 2022) | 2.8B | 65.4 | 35.8 | 19.9 |
| FiD Static (LoRA) | 0.2B | 62.1 | 29.3 | 22.2 |
| FiD Static (LoRA) | 0.8B | 64.5 | 32.9 | 20.4 |

## B.2  IMPACT OF REMOVING NOISY DATA

The significance of the quality of news articles appended to each question is paramount in facilitating accurate predictions. In our empirical investigation, we find that a filtering mechanism, which eliminates less-relevant news articles based on retrieval scores, can result in performance enhancements, even though it may entail a reduction in the overall number of training instances. The score-based thresholding is based on the overall retrieval score defined in Eq. (4), which means we employ both the relevance re-ranking and recency re-ranking. Nevertheless, it is crucial to acknowledge that the inclusion of noisy or irrelevant news articles may have a counterproductive effect, potentially leading to confusion rather than bolstering the model's capacity. This underscores the imperative nature of meticulous pre-processing procedures and the necessity of guiding the model to focus its attention on informative news sources. We perform these comparison experiments based on the vanilla FiD Static (Zou et al., 2022) configurations without changing the retriever or reader module.

Table 5: Effects of removing noisy data on model performance.

| Model | Score Threshold | Model Size | T/F↑ | MCQ↑ | Num↓ |
|---|---|---|---|---|---|
| FiD Static (Zou et al., 2022) | n/a | 0.2B | 62.0 | 29.6 | 24.5 |
| FiD Static (re-run) | 0.0 | 0.2B | 62.1 | 29.3 | 22.2 |
| FiD Static (re-run) | 0.5 | 0.2B | 64.8 | 43.0 | 20.3 |

## B.3 Splitting numerical and non-numerical questions

Numerical questions entail outputting continuous values, which stands in contrast to the discrete choice tasks of T/F and MCQ. Through empirical analysis, we observed that jointly training these three tasks can potentially impede performance in one area or another. However, when we segregate the numerical questions within the training set, we observe a noteworthy enhancement on T/F and MCQ. We conduct the ablation experiments with the relevance re-ranked and summarized news, and keep the same retriever and reader as the vanilla FiD Static (Zou et al., 2022). Please note that we do not conduct numerical question binning in these experiments.

Table 6: Effects of splitting numerical questions on model performance.

| Model | Config | Model Size | T/F↑ | MCQ↑ | Num↓ |
|---|---|---|---|---|---|
| FiD Static (Zou et al., 2022) | - | 0.2B | 62.0 | 29.6 | 24.5 |
| FiD Static (re-run) | rel. re-rank + sum. news | 0.2B | 65.5 | 42.1 | 21.3 |
| FiD Static (re-run) | rel. re-rank + sum. news w/o num. | 0.2B | 65.9 | 43.1 | n/a |
| FiD Static (re-run) | rel. re-rank + sum. news w/ num. only | 0.2B | n/a | n/a | 20.0 |

In comparison to the results in Table 3, we can find that using binning for numerical questions can consistently lead to better performance metrics.

## B.4 Impact of context size

The context size denotes the quantity of news articles we input into our model to facilitate future event forecasting. In an ideal scenario, a greater number of articles can offer more information, potentially enhancing performance. However, this choice also entails the trade-off of increased computational resources and nosier information. To determine the optimal context size, we perform ablation studies with the relevance re-ranked and summarized news, and the outcomes are presented in Table 7. We carry out the ablation experiments using the same retriever and reader network as the vanilla FiD Static (Zou et al., 2022). Indeed, larger context windows do not necessarily enhance performance as they may include less pertinent news articles, potentially leading to confusion in the model's responses.

Table 7: Effects of news article context size on model performance.

| Model | Context | Model Size | T/F↑ | MCQ↑ | Num↓ |
|---|---|---|---|---|---|
| **Small-size Models** | | | | | |
| FiD Static (Zou et al., 2022) | raw news | 0.2B | 62.0 | 29.6 | 24.5 |
| FiD Static (re-run) | rel. re-rank + sum. news, ctx@10 | 0.2B | 65.5 | 42.1 | 21.3 |
| FiD Static (re-run) | rel. re-rank + sum. news, ctx@30 | 0.2B | 66.0 | 42.8 | 20.7 |
| FiD Static (re-run) | rel. re-rank + sum. news, ctx@100 | 0.2B | 65.3 | 40.2 | 21.3 |
| **Middle-size Models** | | | | | |
| FiD Static (Zou et al., 2022) | raw news | 0.8B | 64.1 | 32.4 | 21.8 |
| FiD Static (re-run) | rel. re-rank + sum. news, ctx@10 | 0.8B | 65.8 | 41.0 | 19.8 |
| FiD Static (re-run) | rel. re-rank + sum. news, ctx@30 | 0.8B | 66.4 | 43.4 | 19.9 |
| FiD Static (re-run) | rel. re-rank + sum. news, ctx@100 | 0.8B | 64.2 | 40.1 | 23.3 |

## C ADDITIONAL QUALITATIVE RESULTS

We utilize the pre-trained GPT-3 model (Brown et al., 2020) (`gpt-3.5-turbo` API) for the relevance assessment and text summarization. We show some qualitative results below.

### C.1 QUESTION-NEWS RELEVANCE ASSESSMENT

Intuitively, news articles that are more semantically relevant contribute more to the prediction of future event outcomes. This motivates us to select the most relevant news articles.

---

**Question:** Will Apple release a new iMac based on Apple silicon in 2021? *Choices*: Yes or No. *Start time:* 2021-02-19. *Expiry time:* 2021-05-01.
**News:** *Title:* A new iMac with Apple silicon may be coming even sooner than expected. *Date:* 2021-03-25. *Article:* Credit: IDG We all know an Apple silicon-based iMac is on the way sometime this year, but it might be coming very very soon. After retiring the iMac Pro and discontinuing some higher-end storage configurations of the iMac 4K, the latest macOS Big Sur 11.3 beta contains some clues that new models might be landing very soon. As reported by 9to5Mac, the fifth Big Sur 11.3 beta that landed this week contains references to iMac21,1 and iMac21,2 code identifiers that haven't been seen before. 9to5Mac says they match the Apple silicon codenames previously reported by Bloomberg, though there's nothing out of the ordinary about the numbering. Last year's iMac refreshes were labelled iMac20,1, iMac20,2. But it still points to a possibly imminent release. While the existence of the identifiers doesn't necessarily mean new iMacs are around the corner, it certainly lends some more fuel to that particular fire. A new iMac form factor has been rumoured for more than a year, bringing a Pro XDR Display-inspired redesign that shrinks the bezels and eliminates the chin, and it stands to reason that the next Apple silicon-based Mac will be the iMac. New MacBooks are seemingly slated for the second half of 2021 and the Mac Pro isn't rumoured to arrive until 2022. It's been nearly 10 years since the iMac received its thin-edge design and another eight before it had its last major redesign. The current iMac form factor is basically a variation of the iMac G5 that replaced the iconic iMac G4 in 2004.
**Relevance Assessment:** [4, 4, 4, 4, 4] (We prompt the pre-trained LLM multiple times to provide a relevance score in the range of 0 to 4, with larger numbers indicating higher relevance.)

Figure 4: Example 1 of question-news relevance assessment.

---

**Question:** Will Apple release a new iMac based on Apple silicon in 2021? *Choices*: Yes or No. *Start time:* 2021-02-19. *Expiry time:* 2021-05-01.
**News:** *Title:* The next Apple event is actually happening on April 20 for real. *Date:* 2021-04-14. *Article:* So there is an event after all.Credit: Apple. After rumours, missed predictions, shaved eyebrows, and a surprising Siri leak, we know officially know when Apple's spring event will be held: April 20 at 10 am PT. What will reveal is another question. Apple is calling the event Spring Loaded and the tagline is accompanied by an Apple logo in the shape of a spring. The colours match those in the original Apple rainbow, but it's hard to decipher much of anything else. There's also an AR component that features the coloured lines swirling around the room and converging to form the Apple logo in the invitation. As they have done with other recent events, theres an AR experience on the new #SpringLoaded Apple Event. Visit https://t.co/xfYtZ6OS1v on your iPhone and tap the event image. (Sleeping kitty optional.) pic.twitter.com/9fZrklwEx9 — Jason Cross (@JasonCross00) April 13, 2021 If we're speculating, it could point to the release of a new Apple Pencil alongside a revamped iPad Pro or possibly a new Apple silicon iMac in colourful case options. We're also expecting AppleTags, an updated Apple TV, and AirPods, but speculation has suggested that some of the products may be held until the fall. The event will once again be virtual and streamed on Apple's website and YouTube channel.
**Relevance Assessment:** [3, 2, 3, 3, 1] (We prompt the pre-trained LLM multiple times to provide a relevance score in the range of 0 to 4, with larger numbers indicating higher relevance.)

Figure 5: Example 2 of question-news relevance assessment.

## C.2 TEXT SUMMARIZATION

Below, we showcase additional qualitative results of the pretrained LLM-enhanced text summarization. Typically, the summarization process truncates URL links and less vital details like the report's location. The resulting summarized text is generally logically coherent, significantly more concise, and preserves most of the key information from the original text.

---

**Original Text:** *Title:* Beyond Outperforms US in SMB Processing Growth & Retention. *Article:* PRINCETON, N.J., Sept. 16, 2020 /PRNewswire/ – Beyond outperformed the SMB market in both year over year growth in processing volume and in merchant retention, when compared to a recent report published by top electronic payments analytics and consulting firm The Strawhecker Group (TSG). Beyond increased Net Revenue by 23 percent during Q2, exceeding the aggregate U.S. small and medium business (SMB) market's performance, which had a net revenue reduction of 19 percent during the same time period. Analysis on the U.S. SMB market was conducted using TSG's proprietary Acquiring Industry Metrics (AIM) platform, which tracks more than 45 percent of all U.S. processing merchants and more than one trillion dollars in annualized transactions. "I am so proud of all the hard work our team put in to make this happen," said Beyond founder and CEO Robert O. Carr. "We make promises to both our clients and sales team that we will not arbitrarily raise prices (for clients) or change our compensation structure (for sales). The fact that we could get through the worst of the COVID-19 crisis without breaking these promises, is a testament to the fact that people (whether salespeople or customers) still appreciate being treated fairly and with respect." "We have built the most effective team in the business," said Chief Sales Officer Brent Rose. "We give our sales and service teams the tools to succeed and we provide them with the best support team in the industry." Information about sales careers at Beyond can be found here. About Beyond Beyond is a financial technology and transactions company offering a suite of business tools and services including payment processing, employee management, business capital, and point-of-sale. About The Strawhecker Group The Strawhecker Group (TSG) is a fast-growing analytics and consulting firm. The company serves the entire payments ecosystem, from fintech startups to Fortune 500 companies. The firm provides its clients with advisory services, research and analytics to help them plan and execute their strategic initiatives. Based in Omaha, a recognized payments industry hub, TSG is an established leader in this high-growth, ever-evolving space. [email protected] Sources: https://thestrawgroup.com/

**After Summarization:** Beyond, a financial technology and transactions company, has surpassed the small and medium business (SMB) market in both year-over-year growth in processing volume and merchant retention, according to a report by The Strawhecker Group (TSG). Beyond achieved a 23% increase in Net Revenue during Q2, outperforming the overall U.S. SMB market, which experienced a net revenue reduction of 19% in the same period. The analysis was conducted using TSG's proprietary Acquiring Industry Metrics (AIM) platform, which tracks a significant portion of U.S. processing merchants and transactions. Beyond's success is attributed to its commitment to not raising prices for clients or altering its compensation structure for sales amid the COVID-19 crisis. The company's effective team and support infrastructure have also contributed to its accomplishments. Beyond offers a suite of business tools and services, including payment processing, employee management, business capital, and point-of-sale solutions. The Strawhecker Group (TSG) is a growing analytics and consulting firm that serves the entire payments ecosystem, providing advisory services and research to assist clients in their strategic initiatives.

---

Figure 6: Example 1 of text summarization.

**Original Text:** *Title:* Charles Barkley Said The Bucks Will Win The 2019 NBA Championship. *Article:* Twitter The Milwaukee Bucks put together an impressive Game 4 victory in Boston and, since a disappointing start to the Eastern Conference Semifinals, Mike Budenholzer's team has quieted doubters with improved play. In the immediate aftermath of a 12-point win to take a 3-1 series lead, Milwaukee found itself with a (very) famous advocate, with TNT's Charles Barkley loudly proclaiming that the Bucks "are going to win the world championship." "The @Bucks are going to win the world championship." Sir Charles has spoken. pic.twitter.com/R8hXQN9nYJ — NBA on TNT (@NBAonTNT) May 7, 2019 Of course, it has to be noted that Barkley likes to make bold stances and, as you can hear in the clip above, he predicted that the Portland Trail Blazers would represent the Western Conference in the NBA Finals, even with the Golden State Warriors still alive and kicking. Still, this was definitive from Barkley and, in some ways, it isn't a crazy thought. After all, the Bucks were pretty easily the best team in the NBA during the regular season, racking up 60 wins with a league-leading net rating and dominant units on both ends of the floor. In addition, Giannis Antetokounmpo has played at an MVP level from wire to wire and it isn't wild to say that he would be able to go toe-to-toe with any player in the league today. The Bucks entered Monday evening as the team with the second-best betting odds to win the title and, frankly, that seems right. Barkley isn't one to shy away from a blazing opinion, though, and he laid this one out with perfect clarity.

**After Summarization:** Charles Barkley, a prominent NBA analyst, confidently declared on Twitter that the Milwaukee Bucks would win the 2019 NBA Championship. He made this assertion following the Bucks' strong performance in Game 4 against the Boston Celtics, where they took a 3-1 series lead. Barkley's statement is notable given his tendency to make bold predictions, as he had previously predicted the Portland Trail Blazers to reach the NBA Finals despite the presence of the Golden State Warriors. While his proclamation about the Bucks winning it all might seem audacious, it's not entirely unreasonable. The Bucks had been the dominant team during the regular season with the best record, impressive net rating, and strong performances on both offense and defense. Led by Giannis Antetokounmpo, who was playing at an MVP level, the Bucks were considered among the top contenders for the championship, as reflected in their favorable betting odds. Despite Barkley's penchant for bold opinions, he confidently expressed this prediction based on the Bucks' impressive season and strong roster.

Figure 7: Example 2 of text summarization.

**Original Text:** *Title:* Paytm employee tested positive with coronavirus after returning from Italy. *Article:* A Paytm employee in Haryana's Gurgaon (now officially known as Gurugram) has been tested positive for the novel coronavirus, the company said in a statement on Wednesday. The employee had recently returned from a vacation in Italy. Italy has been emerging as one of the worst-hit countries from coronavirus. The company has advised its employees to work from home for the next couple of days while the Gurugram unit gets sanitised, according to the statement. It also said that as a precautionary measure, they have suggested their team members get health tests done immediately. Meanwhile, the Paytm employee who has been tested positive is receiving appropriate treatment, the company as quoted by news agency ANI. Paytm: One of our colleagues based out of Gurgaon office who recently returned from Italy has sadly been tested positive for Coronavirus. He is receiving appropriate treatment. As a precautionary measure, we have suggested his team members to get health tests done immediately. pic.twitter.com/gXol1a4vOU — ANI (@ANI) March 4, 2020 - article continues after ad - - article resumes - Following today and day before's developments, where a Noida man was tested positive for coronavirus, Paytm, one of India's leading digital payment banks has decided to shut its offices in Gurugram and Noida for at least two days. Today it was reported that the Noida man's (who was recently tested positive after he returned from Italy) six relatives in Agra have been tested positive for COVID-19. He had travelled to Agra to meet them. Morever, after an Italian man and his wife tested positive in Jaipur, today 14 Italian members from the tourist group and their Indian driver also tested positive, making total 17 infections in the group. Yesterday, Shriram Millennium school in Sector 135 Noida decided to shut down the premises for two days and also postpone its annual examinations amid concerns that their student's parent, had tested positive with COVID-19 and has been isolated on Monday. This followed reports that the individual who contracted coronavirus in Italy had attended a birthday party on February 28 (Friday) with his child studying in the Noida school. This party was also attended by other students of the school and their family members. Prior to this, a techie from Hyderabad was suspected to have contracted the virus. With these, the total count of known coronavirus infected cases in India rose to 29, including sixteen Italians. Coronavirus has infected more than 89,000 people globally and caused over 3,000 deaths. The World Health Organization has named the illness COVID-19, referring to its origin late last year and the coronavirus that causes it.

**After Summarization:** An employee of Paytm in Gurgaon, India, tested positive for coronavirus after returning from Italy, a country severely affected by the virus. Paytm advised its employees to work from home temporarily while the office is sanitized. The company also recommended health tests for its team members as a precaution. Paytm's offices in Gurugram and Noida were closed for two days due to a positive case in Noida and related developments. A man in Noida who had returned from Italy tested positive, along with six of his relatives in Agra. Additionally, a group of Italian tourists and their Indian driver tested positive in Jaipur. The virus spread to a Noida school where a parent had attended a party with other students and family members. The total number of confirmed coronavirus cases in India reached 29, including 16 Italians. Globally, COVID-19 has infected over 89,000 people and caused 3,000 deaths. The World Health Organization named the illness COVID-19.

Figure 8: Example 3 of text summarization.

