# OpenReview forum: "AutoCast++: Enhancing World Event Prediction with Zero-shot Ranking-based Context Retrieval"
_ICLR.cc/2024/Conference — ICLR 2024 poster_

### Official Review · Reviewer_vd8q · 2023-10-19

**Soundness:** 3 good
**Presentation:** 2 fair
**Contribution:** 2 fair
**Rating:** 6
**Confidence:** 4

**Summary:**

This paper proposes a zero-shot ranking-based retriever-reader model for event forecasting. The model can generate the effective answers by the proposed task-aligned retrieval module and enhanced neural article reader. Experimental results on a public dataset verify the effectiveness of the proposed method.

**Strengths:**

1.The logic of the paper is sound.
2.The description of the methodology is relatively clear.

**Weaknesses:**

1.The writing of the paper needs to be further improved and all the symbols need to be interpreted.
2.The work focuses on a single textual modality, so why mention multiple data sources, multiple modalities in the introduction section?
3.N_q is a subset of D. Why are the elements in N_q n and not d?
4.The dataset used is too small and results on more as well as larger datasets are needed to be validated.

**Questions:**

See above.

---

> ### Author Response · Authors · 2023-11-22
>
> >**Q1: The writing of the paper needs to be further improved and all the symbols need to be interpreted.**
>
> Thank you for your comment. We have updated the notation explanations in the PDF to enhance clarity.
>
> >**Q2: The work focuses on a single textual modality, so why mention multiple data sources, multiple modalities in the introduction section?**
>
> Thank you for your feedback. In response, we have adjusted our text to more explicitly state our objective of exploring data modalities that are more complex than traditional structured time-series data. The revised sentence is presented below for reference:
>
> *However, as the demand for more accurate forecasts in diverse domains has grown, the need to integrate data from **beyond the structured time-series modality** has become apparent.*
>
>
> >**Q3: N_q is a subset of D. Why are the elements in N_q n and not d?**
>
> Thank you for pointing out this inconsistency. We have updated our notation to use $n$ instead of $d$ to represent each news article, with $n$ serving as an abbreviation for 'news'. Our text has been revised accordingly to reflect this change.
>
> >**Q4: The dataset used is too small and results on more as well as larger datasets are needed to be validated.**
>
> Thank you for your feedback. The Autocast dataset [1] comprises approximately 5700 questions, including both training and testing sets. In comparison, the most similar dataset, ForecastQA [2], contains slightly over 10,000 questions in total. However, we contend that the quantity of questions should not be the primary criterion for assessing the actual scope of such datasets.
>
> While Autocast has fewer questions, it offers a significantly larger context retrieval space. ForecastQA's questions are limited to the time span from January 11, 2019, to November 12, 2019, all within the year 2019. In contrast, Autocast's questions span from 2016 to 2022, necessitating the retrieval and reader module to potentially handle more intricate information that encompasses greater long-term unpredictability.
>
> Furthermore, as highlighted in the Autocast paper [1], the closest benchmark dataset, ForecastQA, has certain limitations in its data. We quote from [1] as follows:
>
> > ForecastQA's questions were authored by crowdworkers without any forecasting experience. Consequently, these questions often lack coherence or clarity due to the absence of additional context, as exemplified by queries like "To how many people will the Representative of an internet speak to by September 2019?" or "In July 2019, will an article say there were no volunteers in 2016?"
>
> We believe that exploring our methodology on additional datasets is a promising avenue for future research, and we are committed to pursuing this in our future work.
>
> Ref:\
> [1] Forecasting future world events with neural networks. NeurIPS 2022.\
> [2] FORECASTQA: A Question Answering Challenge for Event Forecasting with Temporal Text Data. ACL 2021.

---

> > ### Author Response · Authors · 2023-11-23
> >
> > As the rebuttal period is coming to an end, we gently remind you to look over our responses and updated findings. Your attention to our modifications and answers is greatly valued. We sincerely thank you for the time and effort you've put into reviewing our work.

---

> > > ### Comment · Reviewer_vd8q · 2023-11-23
> > > **Response to authors' rebuttal**
> > >
> > > Thanks to the authors for the detailed response. My confusions have been answered and I am willing to revise my rating to "marginally above the acceptance threshold".

---

> > > > ### Author Response · Authors · 2023-11-23
> > > >
> > > > Thanks!

---

### Official Review · Reviewer_mh89 · 2023-10-29

**Soundness:** 2 fair
**Presentation:** 3 good
**Contribution:** 2 fair
**Rating:** 3
**Confidence:** 3

**Summary:**

This paper introduces AutoCast++, an event prediction system designed to address forecasting questions by analyzing news documents. AutoCast++ comprises three key components: the Task-Aligned Retrieval Module, which re-ranks news documents based on relevance (using GPT-3 in a zero-shot manner) and recency; the Enhanced Neural Article Reader, which summarizes pertinent news content; and the Human-Aligned Loss Function, aligning system confidence with human forecaster accuracy. These components collectively led to improvements in addressing various forecasting question types in the AutoCast dataset, notably achieving a 48% enhancement in handling multiple-choice questions.

**Strengths:**

- Ablation study is done to evaluate the effectiveness of different components of the system.

- Their proposed system achieved 48% improvement on multiple choice forecasting questions, which is significant.

- Even the smaller version of their proposed system (with 0.2 and 0.8 billion parameters) outperforms larger baselines (with 2.8 billion parameters).

**Weaknesses:**

- Limited reproducibility: The source code associated with the research paper is unavailable, so it is not possible to reproduce the results.

- In ablation study, no specific experiment has been conducted to demonstrate the isolated effectiveness of the Alignment Loss component.

**Questions:**

Please see the weakness section.

---

> ### Author Response · Authors · 2023-11-22
>
> We are thankful for the reviewer's valuable feedback.
> >**Q1. Limited reproducibility.**
>
> We acknowledge the reviewer's concern regarding reproducibility. In our study, a significant finding is the contribution of pre-trained GPT-3 in enhancing event prediction by performing relevance re-ranking and text summarization. This is conducted on the original datasets that consist of prediction questions and a large number of news articles. During our implementation, these preliminary steps are executed before the training phase. We then save these enhanced retrieval results and integrate them into our dataloader for training the model. To achieve full reproducibility, it is indeed necessary to share the re-ranking results generated by GPT-3, which essentially constitute our modified training data. However, the substantial size of this data exceeds typical file size limitations. As a commitment to transparency and reproducibility, we plan to make these materials publicly available following the acceptance of our paper.
>
> >**Q2. The isolated effectiveness of the Alignment Loss component.**
>
> Please kindly refer to Table 3 in our paper, where we present ablation studies focused on the impact of disabling the alignment loss in our model.
>
> Additionally, in this rebuttal, we trained the FiD Static baseline with additional alignment loss. The summarized results of these experiments, specifically for the 0.2B model sizes, are detailed in the table below.
>
> | Model                             | Retriever (GPT)   | Retriever (non-GPT) | Reader (GPT)  | Reader (non-GPT)  | Reader (non-GPT) | T/F$\\uparrow$ | MCQ$\\uparrow$ | Num$\\downarrow$ |
> |-----------------------------------|-------------------|---------------------|---------------|-------------------|------------------|----------------|----------------|------------------|
> |                                   | Relevance Re-rank | Recency Re-rank     | Summarization | Numerical Binning | Alignment Loss   |                |                |                  |
> |             FiD Static            | No                | No                  | No            | No                | No               | 62.0           | 29.6           | 24.5             |
> |       FiD Static + alignment      | No                | No                  | No            | No                |          **Yes** | 62.5           | 31.2           | 23.5             |
> | FiD Static + GPT3| **Yes**           | No                  | **Yes**       | No                |               No | 65.5           | 42.1           | 21.3             |
> | Autocast++ without alignment loss | Yes               | Yes                 | Yes           | Yes               | **No**           | 66.7           | 43.5           | 19.9             |
> | Autocast++ (ours)                 | Yes               | Yes                 | Yes           | Yes               | Yes              | 66.7           | 43.8           | 19.8             |
>
> This alignment loss is beneficial for the plain FiD Static baseline, which lacks any LLM-enhanced components. However, in the full Autocast++ model, where relevance re-ranking and text summarization are employed, the alignment loss appears to have a diminished role in enhancing performance.
>
> In our additional experiment `FiD Static + GPT3`, we demonstrate that the baseline performance can be significantly enhanced by using the pretrained GPT3-enabled techniques. We acknowledge that the effect of alignment loss is relatively minor when compared to using pretrained LLM in the retriever and reader.

---

> > ### Author Response · Authors · 2023-11-23
> >
> > Following the reviewer's request, we are happy to release our code. Here is an anonymized version of the code for review. We will update the link to be the public (non-anonymous) version after the review process.
> > https://anonymous.4open.science/r/Autocastplusplus-Anonymous-Code

---

> > > ### Author Response · Authors · 2023-11-23
> > >
> > > As the rebuttal period is coming to an end, we gently remind you to look over our responses and updated findings. Your attention to our modifications and answers is greatly valued. We sincerely thank you for the time and effort you've put into reviewing our work.

---

> ### Comment · Area_Chair_zCvD · 2023-12-03
>
> Hi reviewer mh89,
>
> Can you kindly check the authors' rebuttal to the weaknesses you pointed out? Specifically, the authors have responded to
> - The absence of a detailed ablation on the alignment loss in isolation.
> - The lack of open-source code.

---

### Official Review · Reviewer_KQUb · 2023-10-31

**Soundness:** 3 good
**Presentation:** 2 fair
**Contribution:** 3 good
**Rating:** 8
**Confidence:** 4

**Summary:**

The authors present AutoCast++, a system for world event prediction relying on three components: a task-aligned retrieval module; a news summarisation module (text summarisation on retrieved news); a fusion-in-decoder model that is aligned to perform the event predictions.
They evaluate the system on the AutoCast dataset by grouping the tasks in numerical, multiple choice and true/false; considering as baselines a collection of methodologies suggested by the benchmark.
The results show that the proposed system is able to outperform the considered baselines considering different model sizes.

**Strengths:**

The proposed system shows remarkable performance presenting a limited impact from the model size.
The only tasks where it does not excel are the numerical ones, but it's anyway a close call with a baseline that is almost two times larger.

**Weaknesses:**

While the exclusion of baselines relying on new LLMs including data post mid 2021 is understandable, the ablation studies seem to suggest that relying on LLM for retrieval reranking and summarisation play a huge role in the performance of the system.
What would be convincing is to build/revamp the baselines considered using the GPT3 pre-trained version that the authors leverage in their experiments.
This would surely make the submission much stronger and convincing.

**Questions:**

Which GPT3 version was considered in the work?
What is the impact of binning numerical questions? Is the binning applicable also to the baselines? If yes how would results change?

**Details Of Ethics Concerns:**

Relying on LLM-powered systems for world event forecasting using a pre-trained model with no safeguards can be very dangerous in practice. It would be useful to have this discussed in the paper.

---

> ### Author Response · Authors · 2023-11-22
>
> We are grateful for the insightful feedback provided by the reviewer. Below, we address the reviewer's queries and have also updated our manuscript accordingly, with the changes highlighted in blue in the PDF.
>
> >**Q1: revamp the baselines considered using the GPT3.**
>
> We have expanded our experimental results to include the FiD Static (0.2B model size) baseline, now enhanced with retrieval re-ranking and summarization using GPT-3.
>
> | Model                         | Retriever (GPT)   | Retriever (non-GPT) | Reader (GPT)  | Reader (non-GPT)  | Reader (non-GPT) | T/F$\\uparrow$ | MCQ$\\uparrow$ | Num$\\downarrow$ |
> |-------------------------------|-------------------|---------------------|---------------|-------------------|------------------|----------------|----------------|------------------|
> |                               | Relevance Re-rank | Recency Re-rank     | Summarization | Numerical Binning | Alignment Loss   |                |                |                  |
> |           FiD Static          | No                | No                  | No            | No                | No               | 62.0           | 29.6           | 24.5             |
> | FiD Static + GPT3| **Yes**           | No                  | **Yes**       | No                |               No | 65.5           | 42.1           | 21.3             |
> | Autocast++             | **Yes**           | Yes                 | **Yes**       | Yes               | Yes              | 66.7           | 43.8           | 19.8             |
>
> Our findings demonstrate a notable improvement in the FiD Static model when incorporating GPT-3 enhancements in the retriever and reader stages. With these enhancements, the FiD Static baseline now exhibits performance much closer to our model.
>
> It's important to note that the UnifiedQA and T5 baselines are not reliant on retrieval processes (as they do not utilize context from news articles), and thus, are not impacted by this modification.
>
> Regarding the FiD Temporal baseline, it uniquely utilizes daily news articles as context. Given that the average question in the Autocast dataset spans approximately 130 days, this would necessitate processing a volume of articles about 130 times greater than our current setup using the pretrained GPT-3. While this presents an intriguing direction for future research, it was beyond our capacity to fully explore within the limited time frame allocated for this rebuttal.
>
> >**Q2: Which GPT3 version was considered in the work?**
>
> In our experiments, we utilized the gpt-3.5-turbo-0613 API version of GPT-3.
>
> >**Q3: impact of binning numerical questions and results on the baselines?**
>
> Our approach is based on the T5 transformer model [2], and this enables the easy integration of our numerical question binning method with the FiD Static baseline, which is also based on T5. We conducted comparative ablation studies regarding the binning on both the FiD Static and our Autocast++ models, each with a 0.2B model size.
>
> | Model                         | Retriever (GPT)   | Retriever (non-GPT) | Reader (GPT)  | Reader (non-GPT)  | Reader (non-GPT) | T/F$\\uparrow$ | MCQ$\\uparrow$ | Num$\\downarrow$ |
> |-------------------------------|-------------------|---------------------|---------------|-------------------|------------------|----------------|----------------|------------------|
> |                               | Relevance Re-rank | Recency Re-rank     | Summarization | Numerical Binning | Alignment Loss   |                |                |                  |
> |           FiD Static          | No                | No                  | No            | No                | No               | 62.0           | 29.6           | 24.5             |
> |      FiD Static + Binning     | No                | No                  | No            | **Yes**           | No               | 62.1           | 29.6           | 23.2             |
> | Autocast++ without binning    | Yes           | Yes                 | Yes       | **No**                | Yes              | 66.7           | 43.6           | 21.0             |
> | Autocast++ (ours)             | Yes           | Yes                 | Yes       | **Yes**               | Yes              | 66.7           | 43.8           | 19.8             |
>
> This modification transforms the numerical question prediction from a regression task to a classification one, aligning it with the loss used for T/F and MCQ. Our empirical findings show that this adjustment consistently enhances the accuracy of numerical predictions and does not significantly impact T/F and MCQ performance.
>
> Ref:\
> [1] Forecasting future world events with neural networks. NeurIPS 2022.\
> [2] Exploring the limits of transfer learning with a unified text-to-text transformer. JMLR 2020.

---

> > ### Comment · Reviewer_KQUb · 2023-11-22
> > **Response to author rebuttal**
> >
> > I would like to thank the authors for addressing most of my comments and conducting experiements that I believe are improving the quality and the soundness of their work.
> > The experiments show that the framework presented is outperforming the baselines even when equipped with components that are making them more competitive with the presented approach.
> > The only part that I believe deserves to be tackled in an accepted submission is the concerns with the respect to ethics and security and potential strategies to have guardrails in place when predicting events.
> > I'm happy to update my score based on the authors' response.

---

> > > ### Author Response · Authors · 2023-11-23
> > >
> > > Thanks! We thank the reviewer for highlighting the need to consider potential ethical and security concerns. Forecasting systems, especially open-ended ones like ours (in the sense that our work and similar systems can make predictions for any potential even that can be presented as a question), naturally require investigating such concerns.
> > >
> > > Potential ethical concerns include:
> > > - Answering potential unethical/illegal questions (e.g. a the answer to a question such as: "Will drug smuggling arrests in <geographical region> increase to <x> in the next year could lead to crime groups reorganizing their logistics).
> > > - Dataset bias in question answering. The biases present in the article dataset the model is trained on could appear in the answers given. As an example, articles might be biased in their coverage of certain world regions or population groups.
> > > - Adversarial attacks on the model (as are already feasible on LLMs) or more simply exploitable vulnerabilities to how the question is presented. An example of that is that the model, if used in a public policy setting, might have exploitable sensitivities to how exactly the question is formulated. An analogous problem already occurs in real-life when exposing the public to surveys or referendums in which the wording of the question has a measurable impact on the answer.
> > >
> > > Strategies to mitigate these issues include:
> > > - Applying alignment techniques such as those listed in [1] to the LLM component. In general, alignment and robustness training techniques that already exist on LLMs could be applied to our proposed approach.
> > > - Rebalancing the question/article datasets to correct for dataset bias.
> > > - Restricting general availability of the approach, or cautioning it's use.
> > >
> > >
> > >
> > > References:\
> > > [1] Aligning Large Language Models with Human: A Survey, Wang et al.

---

### Meta-Review · Area_Chair_zCvD · 2023-12-10

**Metareview:**

The paper proposes a world event prediction system consisting of three main components: a task-aligned retrieval module, a news summarization module, and a fusion-in-decoder model for event predictions. The evaluation on the AutoCast dataset showcases the system's ability to outperform various baseline methodologies, particularly excelling in multiple-choice forecasting questions.

Strengths: The proposed AutoCast++ system demonstrates remarkable performance across various forecasting tasks, especially excelling in multiple-choice questions. The system's sensitivity to model size is limited (i.e., the system is robust). The ablation study is appreciated for evaluating the effectiveness of different system components.

Weaknesses:
1. Revisiting baselines using the GPT-3 pre-trained version for stronger comparisons.
2. Reviewers also identify issues with the paper's writing quality, questions the relevance of mentioning multiple modalities in the introduction, and emphasizes the need for larger datasets for result validation.

**Justification For Why Not Higher Score:**

Reviews suggest improvements in baseline comparisons, writing, and the need for result validation on larger datasets.

**Justification For Why Not Lower Score:**

Reviews acknowledge the system's remarkable performance and robustness and should be accepted.

---

### Decision · Program_Chairs · 2024-01-16

Accept (poster)